# Multivariate Triangular Quantile Maps
# for Novelty Detection

**Jingjing Wang[1],  Sun Sun[2],  Yaoliang Yu[1]**
University of Waterloo[1],   National Research Council Canada[2]
{jingjing.wang, sun.sun, yaoliang.yu}@uwaterloo.ca

## Abstract

Novelty detection, a fundamental task in machine learning, has drawn a lot of recent attention due to its wide-ranging applications and the rise of neural approaches. In this work, we present a general framework for neural novelty detection that centers around a multivariate extension of the univariate quantile function. Our framework unifies and extends many classical and recent novelty detection algorithms, and opens the way to exploit recent advances in flow-based neural density estimation. We adapt the multiple gradient descent algorithm to obtain the first efficient end-to-end implementation of our framework that is free of tuning hyperparameters. Extensive experiments over a number of real datasets confirm the efficacy of our proposed method against state-of-the-art alternatives.

## 1   Introduction

Novelty detection refers to the fundamental task in machine learning that detects "novel" or "unusual" samples in a data stream. It has wide-ranging applications such as network intrusion detection [14], medical signal processing [17], jet design [19], video surveillance [42, 43], image scene analysis [25, 47], document classification [29, 30], reinforcement learning [39], etc.; see the review articles [7, 31, 32, 41] for more insightful applications.  Over the last two decades or so, many novelty detection algorithms have been proposed and studied in the machine learning field, of which the statistical approach that aims to identify low-density regions of the underlying data distribution has been most popular [e.g. 4, 49, 51, 53]. More recently, new novelty detection algorithms based on deep neural networks [e.g. 1, 9, 11, 18, 26, 40, 44, 46, 48, 56, 58, 59] have drawn a lot of attention as they significantly improve their non-neural counterparts, especially in domains (such as image and video) where complex high-dimensional structures abound.

This work offers a closer look of these recent neural novelty detection algorithms, by making a connection to recent flow-based generative modelling techniques [22]. In §2 we show that the triangular map studied in [22] for neural density estimation serves as a natural extension of the classical univariate quantile function to the multivariate setting. Since density estimation is extremely challenging in high dimensions, recent neural novelty detection algorithms all extract a lower dimensional latent representation, whose probabilistic properties can then by captured by our multivariate triangular quantile map. Based on this observation we propose a general framework for neural novelty detection that includes as special cases many classical approaches such as one-class SVM [49] and support vector data description [53], as well as many recent neural approaches [e.g. 1, 40, 46, 58, 59]. This unified view of neural novelty detection enables us to better understand the similarities and subtle differences of the many existing approaches, and provides some guidance on designing next-generation novelty detection algorithms.

More importantly, our general framework makes it possible to effortlessly plug-in recent flow-based neural density estimators, which have been shown to be surprisingly effective even in moderately high dimensions. Furthermore, centering our framework around the (multivariate) triangular quantile map (TQM) also enables us to unify the two scoring strategies in the literature [34]: we can either threshold

the density function [4, 51] or the (univariate) quantile function [49, 53]. Using the multivariate triangular quantile map, *for the first time we can simultaneously perform both, without incurring any additional cost.* In §3, motivated by the sub-optimality of pre-training we cast our novelty detection framework as multi-objective optimization [35] and apply the multiple gradient descent algorithm [12, 15, 36] for the first time. We present an efficient implementation that learns the TQM consistently, end-to-end and free of tuning hyperparameters. In §4 we perform extensive experiments on a variety of datasets and verify the effectiveness of our framework against state-of-the-art alternatives.

We summarize our main contributions as follows:

- We extend the univariate quantile function to the multivariate setting through increasing triangular maps. This multivariate triangular quantile map may be of independent interest for many other problems involving multivariate probabilistic modelling.

- We present a new framework for neural novelty detection, which unifies and extends many existing approaches including the celebrated one-class SVM and many recent neural ones.

- For the first time we apply the multiple gradient descent algorithm to novelty detection and obtain an efficient end-to-end implementation of our framework that is free of any tuning hyperparameters.

- We perform extensive experiments to compare to existing novelty detection baselines and to confirm the efficacy of our proposed framework.

Our code is available at https://github.com/GinGinWang/MTQ.

## 2 A General Framework for Novelty Detection

In this section we present a general framework for novelty detection. Our framework builds on recent progresses in generative modelling and unifies and extends many existing works.

We follow the standard setup for novelty detection [e.g. 7]: Given $n$ i.i.d. samples $\{\mathbf{X}_1, \ldots, \mathbf{X}_n\}$ from an unknown distribution $P$ over $\mathbb{R}^d$, we want to decide if a new (unseen) sample $\tilde{\mathbf{X}}$ is "novel," i.e. if it is *unlikely* to come from the same distribution $P$. Due to lack of supervision, the notion of "novelty" is not well-defined. Practically, a popular surrogate is to identify the low-density regions of the distribution $P$ [4, 49, 51], as samples from these areas are probabilistically unlikely. For simplicity we assume the underlying distribution $P$ has a density $p$ w.r.t. the Lebesgue measure.

We exploit the following multivariate generalization of the quantile function. Recall that the cumulative distribution function (CDF) $F$ and the quantile function $Q$ of a *univariate* random variable $X$ is defined as:

$$F(x) = \Pr(X \leq x), \qquad Q(u) = F^{-1}(u) := \inf\{x : F(x) \geq u\}.$$

While the CDF can be easily generalized to the multivariate setting, it is not so obvious for the quantile function, as its definition intrinsically relies on the total ordering on the real line. However, following [e.g. 13, 16] we observe that if $U$ follows the uniform distribution over the interval $[0, 1]$, then $Q(U)$ follows the distribution $F$. In other words, the quantile function can be defined as a mapping that pushes the uniform distribution over $[0, 1]$ into the distribution $F$ of interest. This alternative interpretation allows us to extend the quantile function to the multivariate setting. We recall that a mapping $\mathbf{T} = (T_1, \ldots, T_d) : \mathbb{R}^d \to \mathbb{R}^d$ is called triangular if for all $j = 1, \ldots, d$, the $j$-th component $T_j$ depends only on the first $j$ coordinates of the input, and it is called increasing if for all $j$, $T_j$ is increasing w.r.t. the $j$-th coordinate when all other coordinates are fixed. We call $\mathbf{T}$ triangular since its derivative is always a triangular matrix (and vice versa).

**Definition 1 (Triangular Quantile Map (TQM))** *Let $\mathbf{X}$ be a random vector in $\mathbb{R}^d$, and let $\mathbf{U}$ be uniform over the unit hypercube $[0, 1]^d$. We call an **increasing triangular** map $\mathbf{Q} = \mathbf{Q}_{\mathbf{X}} : [0, 1]^d \to \mathbb{R}^d$ the triangular quantile map of $\mathbf{X}$ if $\mathbf{Q}(\mathbf{U}) \sim \mathbf{X}$, where $\sim$ means equality in distribution.*

Note that the TQM $\mathbf{Q}$ is *vector-valued*, unlike the CDF which is always real-valued. The existence and *uniqueness* of $\mathbf{Q}$ follows from results in [5]. Our definition immediately leads to the following quantile change-of-variable formula (cf. the usual change-of-variable formula for densities):

**Proposition 1** *Let $\mathbf{T} : \mathbb{R}^d \to \mathbb{R}^d$ be an increasing triangular map. If $\mathbf{Y} = \mathbf{T}(\mathbf{X})$, then*

$$\mathbf{Q}_{\mathbf{Y}} = \mathbf{T} \circ \mathbf{Q}_{\mathbf{X}}. \tag{1}$$

Practically, eq. (1) allows us to easily stack elementary parameterizations of increasing triangular maps together and still obtain a valid TQM.

To our best knowledge, a similar definition, through conditional univariate quantiles, appeared in a number of works [2, 10, 37, 45], albeit mostly as a theoretical tool. Our definition makes the important triangular structure explicit and amenable to parameterization through deep networks. Needless to say, when $d = 1$, the triangular property is vacuous and our definition reduces to the classical quantile function. For a more comprehensive introduction to triangular maps and its recent rise in machine learning, see [22, 33, 50].

**Remark 1** *A different definition of the multivariate quantile map, based on the theory of optimal transport [54], is discussed in a number of recent works [e.g. 8, 13, 16]:* $\mathbf{Q}$ *is instead constrained to be maximally cyclically monotone, i.e. it is the subdifferential of some convex function. On one hand, this definition is invariant to permutations of the input coordinates while ours is not. On the other hand, our definition is composition friendly (see Proposition 1) hence can easily exploit recent progresses in deep generative models, as we will see shortly. The two definitions coincide with each other only when reduced to the univariate case.*

*We note that the recent work of Inouye and Ravikumar [21] proposed yet another similar definition where* $\mathbf{Q}$ *(termed density destructor there) is only required to be invertible. However, this definition does not lead to a* unique *quantile map and it is less computationally convenient.*

We are now ready to present our general framework for novelty detection. Let $\mathbf{f} : \mathbb{R}^d \to \mathbb{R}^m$ be a feature map and $\mathbf{X}$ a random sample from the unknown density $p$. We propose to learn the density[1] $\mathbf{f}_{\#}p$ of the latent random vector $\mathbf{Z} = \mathbf{f}(\mathbf{X})$ using the approach illustrated in [22]. In details, we learn the feature map $\mathbf{f}$ and the TQM $\mathbf{Q}$ *simultaneously* by minimizing the following objective:

$$\min_{\mathbf{f},\mathbf{Q}} \quad \gamma \mathsf{KL}(\mathbf{f}_{\#}p \| \mathbf{Q}_{\#}q) + \lambda \ell(\mathbf{f}) + \zeta g(\mathbf{Q}), \tag{2}$$

where $g$ embodies some potential constraints on the increasing triangular map $\mathbf{Q}$, $\ell$ is some loss associated with learning the feature map $\mathbf{f}$, $q$ is a fixed reference density (in our case the uniform density over the hypercube $[0,1]^m$), $\zeta, \lambda, \gamma \geq 0$ are regularization constants, and we use the KL-divergence to measure the discrepancy between two densities. Exploiting Proposition 1 we parameterize the TQM as the composition $\mathbf{Q} = \mathbf{T} \circ \mathbf{\Phi}^{-1}$, where $\mathbf{\Phi} = (\Phi, \ldots, \Phi)$ with $\Phi$ the CDF of standard univariate Gaussian and $\mathbf{T} : \mathbb{R}^d \to \mathbb{R}^d$ an increasing triangular map. Note that unlike $\mathbf{Q}$ whose support is constrained to the unit hypercube, there is no constraint on the support of $\mathbf{T}$, hence it is easier to handle the latter computationally.

Once the feature map $\mathbf{f}$ and TQM $\mathbf{Q}$ are estimated (see next section), we can detect novel test samples by either thresholding the density function of the latent variable $\mathbf{Z}$ or thresholding its TQM. In details, the density of $\mathbf{Z} = \mathbf{f}(\mathbf{X}) = \mathbf{Q}(\mathbf{U}) = \mathbf{T}(\mathbf{\Phi}^{-1}(\mathbf{U}))$, using the change-of-variable formula, is

$$p_{\mathbf{Z}}(\mathbf{z}) = 1/|\mathbf{Q}'(\mathbf{Q}^{-1}(\mathbf{z}))| = \tfrac{1}{|\mathbf{T}'(\mathbf{T}^{-1}(\mathbf{z}))|} \cdot \prod_{j=1}^{m} \varphi([\mathbf{T}^{-1}(\mathbf{z})]_j), \quad \text{where} \quad \varphi = \Phi'.$$

Thus, we declare a test sample $\tilde{\mathbf{X}}$ to be "novel" if

$$\log |\mathbf{T}'(\mathbf{T}^{-1}(\mathbf{f}(\tilde{\mathbf{X}})))| + \tfrac{1}{2}\|\mathbf{T}^{-1}(\mathbf{f}(\tilde{\mathbf{X}}))\|_2^2 \geq \tau, \tag{3}$$

where $\tau$ is some chosen threshold. Crucially, since $\mathbf{T}$ is increasing triangular, $\mathbf{T}^{-1}$ and the triangular determinant $|\mathbf{T}'|$ can both be computed very efficiently [22]. The (slight) downside of this density approach is that the scale of an appropriate threshold $\tau$ is usually difficult to guess.

Alternatively, we can declare a test sample $\tilde{\mathbf{X}}$ to be "novel" by directly thresholding the TQM $\mathbf{Q}$. Indeed, let $N \subseteq [0,1]^m$ be a subset whose (uniform) measure is $1 - \alpha$ for some $\alpha \in (0,1)$, then we say $\tilde{\mathbf{X}}$ is "novel" iff

$$\mathbf{Q}^{-1}(\mathbf{f}(\tilde{\mathbf{X}})) \notin N. \tag{4}$$

For instance, we can choose $N$ to be the cube centered at $(1/2, \ldots, 1/2)$ and with side length $(1-\alpha)^{1/m}$, in which case

$$\mathbf{Q}^{-1}(\mathbf{f}(\tilde{\mathbf{X}})) \notin N \iff \|\mathbf{Q}^{-1}(\mathbf{f}(\tilde{\mathbf{X}})) - \tfrac{1}{2}\|_\infty \geq (1-\alpha)^{1/m}/2.$$

The upside of this quantile approach is that we can control Type-I error (i.e. false positive) precisely, i.e. if $\tilde{\mathbf{X}}$ is indeed sampled from $p$, then we will declare it to be novel with probability at most $\alpha$.

Before proceeding to the implementation details of (2), let us mention the advantages of our general framework (2) for novelty detection: (a) It allows us to perform feature extraction on the original sample $\mathbf{X}$ in an end-to-end fashion. As is well-known, density estimation hence also novelty detection becomes extremely challenging when the dimension $d$ is high. Our framework alleviates this curse-of-dimensionality by setting $m \ll d$ and employing $\mathbf{f}$ to perform dimensionality reduction. (b) Our end-to-end framework enables us to adopt the recent flow-based density estimation algorithms, which have been shown to be universally consistent [20, 22] and extremely effective in practice. (c) By estimating the TQM $\mathbf{Q}$ once, we can employ the two scoring rules, i.e. the density scoring rule (3) and the quantile scoring rule (4), simultaneously, without incurring any extra overhead. This allows us to perform a fair and comprehensive experimental comparison of the two complementary approaches. (d) Last but not least, our framework recovers, unifies, and extends many existing approaches in the literature. Let us conclude this section with some examples.

**Example 1 (One-class SVM [49])** *As shown in [52], the one-class SVM minimizes precisely the conditional value-at-risk, which is the average of the tail of a distribution:*

$$\min_{f} \ \mathtt{CVaR}_\alpha(f(\mathbf{X})) + \lambda \|f\|_{\mathcal{H}_\kappa}^2, \quad \textit{where} \quad \mathtt{CVaR}_\alpha(Z) := \mathsf{E}(Z | Z \geq Q_Z(\alpha)),$$

*$Q_Z(\alpha)$ is the $\alpha$-th quantile of the real random variable $Z$, and $\mathcal{H}_\kappa$ is the reproducing kernel Hilbert space (RKHS) induced by some kernel $\kappa$. This approach employs the quantile scoring rule (4).*

*To cast one-class SVM into our framework (2), let us set $m = 1$ hence the TQM reduces to the classical one. Let $\ell(f) = \|f\|_{\mathcal{H}_\kappa}^2$ and $g(\mathbf{Q}) = \mathtt{CVaR}_\alpha(\mathbf{Q}_\# q)$. Now with $\zeta = 1$ and $\gamma = \infty$ in (2) we recover the celebrated one-class SVM.*

*If instead of choosing $f$ from an RKHS, we represent $f$ using a deep network, then we recover the recent approach in [6].*

**Example 2 (Support Vector Data Description (SVDD) [53])** *Similar to one-class SVM, it is easy to show that SVDD also minimizes the conditional value-at-risk:*

$$\min_{\mathbf{c} \in \mathcal{H}_\kappa} \ \mathtt{CVaR}_\alpha(\|\boldsymbol{\varphi}(\mathbf{X}) - \mathbf{c}\|_{\mathcal{H}_\kappa}^2),$$

*where $\boldsymbol{\varphi} : \mathbb{R}^d \to \mathcal{H}_\kappa$ is the canonical feature map of the RKHS. This approach also employs the quantile scoring rule (4). It is well-known known that SVDD and one-class SVM are equivalent for radial kernels [e.g. 49].*

*Again in this case $m = 1$. Let $f(\mathbf{X}) = \|\boldsymbol{\varphi}(\mathbf{X}) - \mathbf{c}\|_{\mathcal{H}_\kappa}^2$, $\ell \equiv 0$ and $g(\mathbf{Q}) = \mathtt{CVaR}_\alpha(\mathbf{Q}_\# q)$. As $\gamma$ approaches $\infty$ in (2), we recover the SVDD formulation.*

*If instead of choosing $\boldsymbol{\varphi}$ as the canonical feature map of an RKHS, we represent $\boldsymbol{\varphi}$ using a deep network, then we recover the recent approach in [44].*

**Example 3 (Latent Space Autoregression (LSA) [1])** *The recent work [1], following a sequence of previous attempts [40, 46, 58, 59], proposed to learn the feature map $\mathbf{f}$ using an auto-encoder structure, and to learn the density of the latent variable $\mathbf{Z} = \mathbf{f}(\mathbf{X})$ using an autoregressive model, which, as argued in [22], exactly corresponds to a triangular map. In other words, if we set $\mathbf{f}$ as the parameters of an auto-encoder, $\ell$ to be its reconstruction loss, and $g \equiv 0$, then our framework (2) reduces to LSA. However, our general framework opens the way to exploit more advanced flow-based density estimation algorithms, as well as the quantile scoring rule (4).*

## 3 Estimating TQM Using Deep Networks

In this section we show how to estimate the TQM $\mathbf{Q}$ in (2) based on samples $\{\mathbf{X}_1, \ldots, \mathbf{X}_n\} \overset{i.i.d.}{\sim} p$. In particular, any flow-based neural density estimator can be plugged into our framework.

Our framework (2) has three components which we implement as follows:

- A feature extractor $\mathbf{f}$ for performing dimensionality reduction. Following previous works [1, 40, 46, 58, 59] we implement $\mathbf{f}$ through a deep autoencoder that consists of one encoder $\mathbf{Z} = \mathcal{E}(\mathbf{X}; \boldsymbol{\theta}_E)$

and one decoder $\hat{\mathbf{X}} = \mathcal{D}(\mathbf{Z}; \boldsymbol{\theta}_D)$. We use the Euclidean reconstruction loss:

$$\ell(\mathbf{f}) = \ell(\boldsymbol{\theta}_E, \boldsymbol{\theta}_D) = \sum_{i=1}^{n} \|\mathbf{X}_i - \hat{\mathbf{X}}_i\|^2.$$

As argued in [3], the reconstruction error, aside from low likelihood, is an important indicator for "novelty." Indeed, since the autoencoder is trained on nominal data, a test sample will incur a large reconstruction error only when it is novel, as such samples have never been encountered before.

- A flow-based neural density estimator for $\mathbf{Q}$. Here we adopt the sum-of-squares (SOS) flow proposed in [22], although other neural density estimators would apply equally well. The SOS flow consists of two parts: an increasing (univariate) polynomial $\mathfrak{P}_{2r+1}(u; \mathbf{a})$ with degree $2r+1$ for modelling conditional densities and a conditioner network $C_j(u_1, \ldots, u_{j-1}; \boldsymbol{\theta}_Q)$ for generating the coefficients $\mathbf{a}$ of the polynomial:

$$\mathfrak{P}_{2r+1}(u; \mathbf{a}) = c + \int_0^u \sum_{s=1}^{k} \left( \sum_{l=0}^{r} a_{l,s} t^l \right)^2 \mathrm{d}t,$$

where $c \in \mathbb{R}$ is an arbitrary constant, $r \in \mathbb{N}$ is the degree of polynomial, and $k$ can be chosen as small as 2. In other words, the TQM $\mathbf{Q}$ learned using SOS flow has the following form:

$$\mathbf{Q} = \mathbf{T} \circ \boldsymbol{\Phi}^{-1}, \quad \text{where} \quad \forall j, \; T_j(u_1, \ldots, u_j) = \mathfrak{P}_{2r+1}\big(u_j; C_j(u_1, \ldots, u_{j-1}; \boldsymbol{\theta}_Q)\big). \quad (5)$$

Any regularization term on the conditioner network weights $\boldsymbol{\theta}_Q$ can be put into the function $g(\mathbf{Q})$ in our framework (2).

- Lastly, the KL-divergence term in (2) can be approximated empirically using the given sample $\{\mathbf{X}_1, \ldots, \mathbf{X}_n\}$. Upon dropping irrelevant constants we reduce the KL term in (2) to:

$$\min_{\boldsymbol{\theta}_Q} \quad \sum_{i=1}^{n} \Big[ \log |\mathbf{Q}'(\mathbf{Q}^{-1}(\mathbf{f}(\mathbf{X}_i)))| - \log q(\mathbf{Q}^{-1}(\mathbf{f}(\mathbf{X}_i))) \Big],$$

where each component of $\mathbf{Q}$ is given in (5). Crucially, since $\mathbf{Q}$ is increasing triangular, evaluating the inverse $\mathbf{Q}^{-1}$ and the Jacobian $|\mathbf{Q}'|$ can both be done in linear time [22].

Since $q$ is the uniform density over the hypercube, upon simplification the final training objective we use in our experiments is as follows. Let $\mathbf{Z}_i = \mathcal{E}(\mathbf{X}_i; \boldsymbol{\theta}_E)$, we aim to solve:

$$\min_{\boldsymbol{\theta}} \quad \sum_{i=1}^{n} (1 - \lambda) \Big[ \underbrace{\log |\mathbf{T}'(\mathbf{T}^{-1}(\mathbf{Z}_i))| + \|\mathbf{T}^{-1}(\mathbf{Z}_i)\|_2^2 / 2}_{\text{negative log-likelihood } h(\mathbf{X}_i; \boldsymbol{\theta})} \Big] + \lambda \underbrace{\|\mathbf{X}_i - \mathcal{D}(\mathbf{Z}_i; \boldsymbol{\theta}_D)\|^2}_{\text{reconstruction loss } \ell(\mathbf{X}_i; \boldsymbol{\theta})}, \quad (6)$$

and recall that $\mathbf{Q} = \mathbf{T} \circ \boldsymbol{\Phi}^{-1}$ is parameterized through the conditioner network weights $\boldsymbol{\theta}_Q$ in (5). We did not find it necessary to further regularize $\mathbf{Q}$ hence set $g \equiv 0$ in (2) and w.l.o.g. $\gamma = 1 - \lambda$.

The first KL term in (2), as is well-known, reduces to the negative log-likelihood of the latent random vectors $\mathbf{Z}_i$ in (6), and the second term is the standard reconstruction loss. The two terms share the encoder weights $\boldsymbol{\theta}_E$ and the trade-off is balanced through the hyperparameter $\lambda$. This design choice conforms to the psychology findings in [3]. In practice, we found that the variance of the log-likelihood is much larger than that of the reconstruction loss, and as a consequence we observed substantial difficulty in directly minimizing the weighted objective in (6). A popular pre-training heuristic is to train the whole model in two stages: we first minimize the reconstruction loss $\ell(\boldsymbol{\theta}_E, \boldsymbol{\theta}_D)$ and then, with the learned hidden vector $\mathbf{Z}$, we estimate the TQM $\mathbf{Q}$ by maximum likelihood. However, as shown in [59], the latent representation learned in the first stage does not necessarily help the task in the second stage.

Instead, we cast the two competing objectives in (6) as multi-objective optimization, which we solve using the multiple gradient descent algorithm (MGDA) [12, 15, 36]. Our motivation comes from the following observation: the two-stage procedure amounts to first setting $\lambda = 1$ and running gradient descent (GD) for a number of iterations, then switching to $\lambda = 0$ (or $\lambda = 0.5$ say) and running GD for the remaining iterations. Naturally, instead of any pre-determined schedule for the hyperparameter $\lambda$ (such as switching from 1 to 0 or 0.5), why not let GD decide what $\lambda$ to use in each iteration? This is precisely the main idea behind MGDA, where at iteration $t$ we solve

$$\lambda_t = \operatorname*{argmin}_{0 \le \lambda \le 1} \left\| \sum_{i \in I} (1 - \lambda) \nabla h(\mathbf{X}_i; \boldsymbol{\theta}_t) + \lambda \nabla \ell(\mathbf{X}_i; \boldsymbol{\theta}_t) \right\|^2 = \min \left\{ 1, \max \left\{ 0, \frac{\langle \nabla h_I - \nabla \ell_I, \nabla h_I \rangle}{\|\nabla h_I - \nabla \ell_I\|^2} \right\} \right\},$$

where $I \subseteq \{1, \ldots, n\}$ is a minibatch of samples, and obviously $\nabla h_I = \sum_{i \in I} \nabla h(\mathbf{X}_i; \boldsymbol{\theta}_t)$ and similarly for $\nabla \ell_I$. With $\lambda_t$ calculated we can continue the gradient update:

$$\boldsymbol{\theta}_{t+1} = \boldsymbol{\theta}_t - \eta[(1 - \lambda_t)\nabla h_I + \lambda_t \nabla \ell_I],$$

where $\eta \geq 0$ is the step size. As shown in [12], this algorithm converges to a Pareto-optimal solution under fairly general conditions. Pleasantly, MGDA eliminates the need of tuning the hyperparameter $\lambda$ as it is determined automatically on the fly. To our best knowledge, our work is the first to demonstrate the effectiveness of MGDA on novelty detection tasks.

We end our discussion by pointing out that the algorithm we develop here can easily be adapted to other design choices that fit into our general framework (2). For instance, if we use a variational autoencoder [23] or a denoising autoencoder [55], then we need only replace the square reconstruction loss in (6) accordingly.

## 4 Empirical Results

In this section, we evaluate the performance of our proposed method for novelty detection and compare it with the traditional and state-of-the-art alternatives. For evaluation, we use precision, recall, F1 score, and the Area Under Receiver Operating Characteristic (AUROC) curve as our performance metrics, which are commonly used in previous works.

### 4.1 Datasets

In our experiments, we use two public image datasets: MNIST and Fashion-MNIST, as well as two non-image datasets: KDDCUP and Thyroid. A detailed description of these datasets, the applied network architectures, and the training hyperparameters can be found in Appendix A. For MNIST and Fashion-MNIST, each of the ten classes is deemed as the nominal class while the rest of the nine classes are deemed as the novel class. We use the standard training and test splits. For every class, we hold out $10\%$ of the training set as the validation set, which is used to tune hyperparameters and to monitor the training process.

### 4.2 Competitor Algorithms

We compare our method with the following alternative algorithms:

- **OC-SVM** [49]. OC-SVM is a traditional kernel-based quantile approach which has been widely used in practice for novelty detection. We use the RBF kernel in our experiments. We consider two OC-SVM-based methods for comparison. 1) RAW-OC-SVM: the input is directly fed to OC-SVM; 2) CAE-OC-SVM: a convolutional autoencoder is first applied to the input data for dimensionality reduction, and then the low-dimensional latent representation is fed to OC-SVM.

- **Geometric transformation (GT)** [18]. A self-labeled multi-class dataset is first created by applying a set of geometric transformations to the original nominal examples. Then, a multi-class classifier is trained to discriminate the geometric transformations of each nominal example. The scoring function in GT is the conditional probability of the softmax responses of the classifier given the geometric transformations.

- **Variational autoencoder (VAE)** [23]. The evidence lower bound is used as the scoring function.

- **Denoising autoencoder (DAE)** [55]. The reconstruction error is used as the scoring function.

- **Deep structured energy-based models (DSEBM)** [58]. DSEBM employs a deterministic deep neural network to output the energy function (i.e., negative log-likelihood), which is used to form the density of nominal data. The network is trained by score matching in a way similar to training DAE. Two scoring functions based on reconstruction error and energy score are considered.

- **Deep autoencoding Gaussian mixture model (DAGMM)** [59]. DAGMM consists of a compression network implemented using a deep autoencoder and a Gaussian mixture estimation network that outputs the joint density of the latent representations and some reconstruction features from the autoencoder. The energy function is used as the scoring function.

- **Generative probabilistic novelty detection (GPND)** [40]. GPND, based on adversarial autoencoders, employs an extra adversarial loss to impose priors on the output distribution. The density is

Table 1: AUROC of Variants of Our Method on MNIST

| Scoring function | $\lambda = 0.99$ | 0.9 | 0.5 | 0.1 | Optimized |
|---|---|---|---|---|---|
| NLL | **0.9729** | 0.9692 | 0.9537 | 0.9389 | 0.9728 |
| TQM$_1$ | 0.9622 | 0.9616 | 0.9430 | 0.9319 | **0.9666** |
| TQM$_2$ | 0.9666 | 0.9645 | 0.9465 | 0.9347 | **0.9699** |
| TQM$_\infty$ | 0.9499 | 0.9527 | 0.9371 | 0.9128 | **0.9531** |

Table 2: Average Precision, Recall, and F1 Score on Non-image Datasets

| Method | Thyroid | | | KDDCUP | | |
|---|---|---|---|---|---|---|
| | Precision | Recall | F1 | Precision | Recall | F1 |
| RAW-OC-SVM * | 0.3639 | 0.4239 | 0.3887 | 0.7457 | 0.8523 | 0.7954 |
| DSEBM * | 0.0404 | 0.0403 | 0.0403 | 0.7369 | 0.7477 | 0.7423 |
| DAGMM * | 0.4766 | 0.4834 | 0.4782 | 0.9297 | 0.9442 | 0.9369 |
| Ours-REC | – | – | – | 0.6305 | 0.6287 | 0.6296 |
| Ours-NLL | 0.7312 | 0.7312 | 0.7312 | **0.9622** | **0.9622** | **0.9622** |
| Ours-TQM$_1$ | 0.5269 | 0.5269 | 0.5269 | 0.9621 | 0.9621 | 0.9621 |
| Ours-TQM$_2$ | 0.5806 | 0.5806 | 0.5806 | **0.9622** | **0.9622** | **0.9622** |
| Ours-TQM$_\infty$ | **0.7527** | **0.7527** | **0.7527** | **0.9622** | **0.9622** | **0.9622** |

used as the scoring function. By linearizing the manifold that nominal data resides on, its density is factorized into two product terms, which are then approximately computed using nominal data.

- **Latent space autoregression (LSA)** [1]. A parametric autoregressive model is used to estimate the density of the latent representation generated by a deep autoencoder, where the conditional probability densities are modeled as multinomials over quantized latent representations. The sum of the normalized reconstruction error and log-likelihood is used as the scoring function.

## 4.3 Variants of Our Method

In this subsection, we first compare some variants of our proposed method. With regard to the network configuration, except on Thyroid whose dimension is too small to require any form of dimentionality reduction, all other experiments contain both an autoencoder and an estimation network.

We consider the following five scoring functions that we threshold at some level $\tau$. In particular, given a test example $\tilde{\mathbf{X}}$, we denote its reconstruction by $\hat{\mathbf{X}}$ and its latent representation by $\tilde{\mathbf{Z}} = \mathbf{f}(\tilde{\mathbf{X}})$.

- Reconstruction error (REC): $\|\tilde{\mathbf{X}} - \hat{\mathbf{X}}\|^2$;
- Negative log-likelihood (NLL): $\log|\mathbf{T}'(\mathbf{T}^{-1}(\tilde{\mathbf{Z}}))| + \|\mathbf{T}^{-1}(\tilde{\mathbf{Z}})\|_2^2/2$;
- 1-norm of quantile (TQM$_1$): $\|\boldsymbol{\Phi}(\mathbf{T}^{-1}(\tilde{\mathbf{Z}})) - \frac{1}{2}\|_1$,
- 2-norm of quantile (TQM$_2$): $\|\boldsymbol{\Phi}(\mathbf{T}^{-1}(\tilde{\mathbf{Z}})) - \frac{1}{2}\|_2$;
- Infinity norm of quantile (TQM$_\infty$): $\|\boldsymbol{\Phi}(\mathbf{T}^{-1}(\tilde{\mathbf{Z}})) - \frac{1}{2}\|_\infty$.

In Table 1, we compare two approaches on MNIST for selecting the hyperparameter $\lambda$ in the training phase: 1) chosen from a pre-set family using the validation set; and 2) automatically optimized using MGDA [12, 15, 36]. We report the average AUROC over 10 classes. It is clear that for all scoring functions, the optimized $\lambda$ generally leads to the highest AUROC. This is also observed on other datasets such as Fashion-MNIST. Within the proposed variants, NLL results in the highest AUROC among all scoring functions, followed by TQM$_2$. In Table 2, on the two non-image datasets we evaluate the average precision, recall, and F1 score. The superscript $*$ on the baselines indicates that the results are directly quoted from the respective references. The threshold is chosen by assuming the prior knowledge of the ratio between the novel and nominal examples in the test set. Under this assumption, the number of false positives is equal to that of false negatives, thus the value of the three metrics coincides. On Thyroid, TQM$_\infty$ is slightly better than the density-based method. On KDDCUP, the density and quantile-based approaches have the same performance, while REC results in the worst performance. On both datasets, our proposed methods are superior to the benchmarks.

Table 3: AUROC on MNIST and Fashion-MNIST

|  | MNIST | | | | | | | | | | |
|---|---|---|---|---|---|---|---|---|---|---|---|
| Class | `OC-SVM` | | `VAE` | `DAE` | `LSA` | `GT` | `DAGMM` | `GPND` | `DSEBM` | Ours-`NLL` | Ours-`TQM`$_2$ |
|  | `RAW` | `CAE` | | | | | | | | | |
| 0 | 0.995 | 0.990 | 0.985 | 0.982 | 0.998 | 0.982 | 0.500 | 0.999 | 0.320 | 0.995 | 0.993 |
| 1 | 0.999 | 0.999 | 0.997 | 0.998 | 0.999 | 0.893 | 0.766 | 0.999 | 0.987 | 0.998 | 0.997 |
| 2 | 0.926 | 0.919 | 0.943 | 0.936 | 0.923 | 0.993 | 0.326 | 0.980 | 0.482 | 0.953 | 0.948 |
| 3 | 0.936 | 0.939 | 0.916 | 0.929 | 0.974 | 0.987 | 0.319 | 0.968 | 0.753 | 0.963 | 0.957 |
| 4 | 0.967 | 0.946 | 0.945 | 0.940 | 0.955 | 0.993 | 0.368 | 0.980 | 0.696 | 0.966 | 0.963 |
| 5 | 0.955 | 0.936 | 0.929 | 0.928 | 0.966 | 0.994 | 0.490 | 0.987 | 0.727 | 0.962 | 0.960 |
| 6 | 0.987 | 0.979 | 0.977 | 0.982 | 0.992 | 0.999 | 0.515 | 0.998 | 0.954 | 0.992 | 0.990 |
| 7 | 0.966 | 0.951 | 0.975 | 0.971 | 0.969 | 0.966 | 0.500 | 0.988 | 0.911 | 0.969 | 0.966 |
| 8 | 0.903 | 0.896 | 0.864 | 0.857 | 0.935 | 0.974 | 0.467 | 0.929 | 0.536 | 0.955 | 0.951 |
| 9 | 0.962 | 0.960 | 0.967 | 0.974 | 0.969 | 0.993 | 0.813 | 0.993 | 0.905 | 0.977 | 0.976 |
| avg | 0.960 | 0.952 | 0.950 | 0.950 | 0.968 | 0.977 | 0.508 | **0.982** | 0.727 | 0.973 | 0.970 |
|  | Fashion-MNIST | | | | | | | | | | |
| Class | `OC-SVM` | | `VAE` | `DAE` | `LSA` | `GT` | `DAGMM` | `GPND` | `DSEBM` | Ours-`NLL` | Ours-`TQM`$_2$ |
|  | `RAW` | `CAE` | | | | | | | | | |
| 0 | 0.919 | 0.908 | 0.874 | 0.867 | 0.916 | 0.903 | 0.303 | 0.917 | 0.891 | 0.922 | 0.917 |
| 1 | 0.990 | 0.987 | 0.977 | 0.978 | 0.983 | 0.993 | 0.311 | 0.983 | 0.560 | 0.958 | 0.950 |
| 2 | 0.894 | 0.884 | 0.816 | 0.808 | 0.878 | 0.927 | 0.475 | 0.878 | 0.861 | 0.899 | 0.899 |
| 3 | 0.942 | 0.911 | 0.912 | 0.914 | 0.923 | 0.906 | 0.481 | 0.945 | 0.903 | 0.930 | 0.925 |
| 4 | 0.907 | 0.913 | 0.872 | 0.865 | 0.897 | 0.907 | 0.499 | 0.906 | 0.884 | 0.922 | 0.921 |
| 5 | 0.918 | 0.865 | 0.916 | 0.921 | 0.907 | 0.954 | 0.413 | 0.924 | 0.859 | 0.894 | 0.884 |
| 6 | 0.834 | 0.820 | 0.738 | 0.738 | 0.841 | 0.832 | 0.420 | 0.785 | 0.782 | 0.844 | 0.838 |
| 7 | 0.988 | 0.984 | 0.976 | 0.977 | 0.977 | 0.981 | 0.374 | 0.984 | 0.981 | 0.980 | 0.972 |
| 8 | 0.903 | 0.877 | 0.795 | 0.782 | 0.910 | 0.976 | 0.518 | 0.916 | 0.865 | 0.945 | 0.943 |
| 9 | 0.982 | 0.955 | 0.965 | 0.963 | 0.984 | 0.994 | 0.378 | 0.876 | 0.967 | 0.983 | 0.983 |
| avg | 0.928 | 0.910 | 0.884 | 0.881 | 0.922 | **0.937** | 0.472 | 0.911 | 0.855 | 0.928 | 0.923 |

## 4.4 Comparison with Baseline Methods

In this section, we compare our method with the baseline approaches. Note that except `RAW-OC-SVM` and `GT`, all other methods, including our own, are based on autoencoders.

In Table 3, we show the comparison of AUROC on the image datasets. Among the proposed quantile scoring functions we only list `TQM`$_2$, which outputs the highest value of AUROC. We observe that on both datasets our proposed methods are superior to most of the benchmarks, with the density scoring function being slightly better than the quantile one. On MNIST, `GPND` and `GT` have better performance; and on Fashion-MNIST, `GT` outputs the highest value of AUROC followed by Ours-`NLL` and `RAW-OC-SVM`. However, since `GT` explicitly extracts features by using a set of geometric transformations, it inevitably suffers a high computational and space complexity. In Appendix B, we further compare and discuss the proposed density and quantile-based approaches in detail.

## 4.5 Comparison with Two-Stage Training

In our proposed algorithm the autoencoder and the estimation network are trained jointly by employing MGDA. For comparison, we also consider the following two-stage training strategies:

- We first train the autoencoder, then fix the autoencoder and train the estimation network alone (denoted as Fix-).
- we first pretrain the autoencoder, then jointly train the autoencoder and the estimation network with the weight $\lambda$ fixed to $0.5$ (denoted as Pretrain-).

The comparison regarding AUROC on MNIST is shown in Table 4. We found that the proposed joint training method leads to the best performance for both the density-based and the quantile-based scoring functions. This is consistent with the findings in many existing works [e.g. 1, 6, 44, 58]. For the fixed two-stage method, our understanding is that the latent representation learned in the first stage may not be the most beneficial for the training of the estimation network in the second stage, which in turn degrades the overall performance. For the pretrained two-stage method, although in the second stage the two parts are trained jointly the autoencoder is initialized with the parameters

Table 4: Comparison between joint and two-stage training: AUROC on MNIST

| Class | Fix-NLL | Pretrain-NLL | Ours-NLL | Fix-TQM$_2$ | Pretrain-TQM$_2$ | Ours-TQM$_2$ |
|---|---|---|---|---|---|---|
| 0 | 0.9939 | **0.9954** | 0.9951 | 0.9904 | **0.9939** | 0.9925 |
| 1 | 0.9971 | **0.9988** | 0.9977 | 0.9972 | **0.9985** | 0.9969 |
| 2 | 0.9403 | **0.9677** | 0.9526 | 0.9188 | **0.9568** | 0.9479 |
| 3 | 0.9568 | 0.9496 | **0.9627** | **0.9481** | 0.9414 | 0.9567 |
| 4 | **0.9703** | 0.9445 | 0.9657 | **0.9700** | 0.9388 | 0.9625 |
| 5 | 0.9612 | 0.9564 | **0.9618** | 0.9525 | 0.9486 | **0.9601** |
| 6 | 0.9878 | 0.9907 | **0.9915** | 0.9841 | 0.9881 | **0.9895** |
| 7 | 0.9629 | 0.9676 | **0.9686** | 0.9587 | 0.9656 | **0.9660** |
| 8 | 0.9549 | **0.9587** | 0.9551 | 0.9397 | **0.9527** | 0.9512 |
| 9 | 0.9736 | 0.9733 | **0.9768** | 0.9742 | 0.9641 | **0.9756** |
| avg | 0.9699 | 0.9703 | **0.9728** | 0.9634 | 0.9649 | **0.9699** |

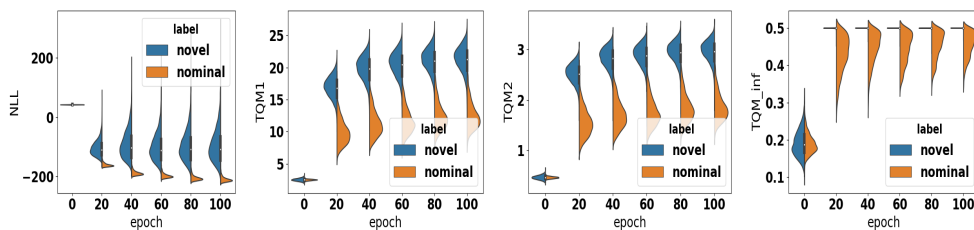

Figure 1: Distributional comparison on training and test scoring statistics on MNIST (nominal: digit 1). From left to right: 1) NLL; 2) TQM$_1$; 3) TQM$_2$; and 4) TQM$_\infty$.

learned in the first stage, which might prevent it from being updated to a more suitable local optimum. The comparison on Fashion-MNIST dataset is similar and is shown in Appendix C.

### 4.6 Visualization

In Figure 1, we show the violin plots of the scoring statistics NLL, TQM$_1$, TQM$_2$, and TQM$_\infty$ on MNIST test set (with digit 1 serving the nominal class). We use the network parameters produced at every 20 epochs in training to generate each curve. We can see that, in the beginning the nominal and novel data have a large region of overlap and after more training epochs they are gradually separated. After about 20 epochs of training they can be clearly distinguished under NLL, TQM$_1$, and TQM$_2$, which indicates the effectiveness of these scoring functions. For TQM$_\infty$, the distribution of novel data is concentrated within a narrow region, which is near the boundary of that of nominal data. More results on visualization can be found in Appendix D.

## 5 Conclusion

The univariate quantile function was extended to the multivariate setting through increasing triangular maps, which in turn motivates us to develop a general framework for neural novelty detection. Our framework unifies and extends many existing algorithms in novelty detection. We adapted the multiple gradient algorithm to obtain an efficient, end-to-end implementation of our framework that is free of any tuning hyperparameters. We performed extensive experiments on a number of datasets to confirm the competitiveness of our method against state-of-the-art alternatives. In the future we will study the consistency of our estimation algorithm for the multivariate triangular quantile map and we plan to apply it to other multivariate probabilistic modelling tasks.

## Acknowledgement

We thank the reviewers for their constructive comments. We thank Priyank Jaini for bringing Decurninge's work to our attention. This work is supported by NSERC.

## Footnotes

[1] The notation $\mathbf{T}_{\#}p$ stands for the push-forward density, i.e., the density of $\mathbf{T}(\mathbf{X})$ when $\mathbf{X} \sim p$.

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
