[Supplementary Material]

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

# A  Datasets and network architectures

In this section, we briefly describe the datasets, the network architectures, as well as the hyperparameters that are used in our proposed algorithm. For all image datasets, the pixel values of each image are scaled to $[0, 1]$. For non-image datasets, no extra preprocessing is applied. The statistics of the datasets are summarized in Table 5.

- **MNIST** [24]
    - Dataset description: MNIST [24] includes $70,000$ grayscale images of numeric digits from 0 to 9, each of size $28 \times 28$. There are $7,000$ examples per class. The training set contains $60,000$ examples, and the test set contains $10,000$ examples.
    - Network architecture: We use the same autoencoder as that in LSA, and the dimension of the latent vector is set to $64$. The estimation network is based on SOS [22], which contains multiple blocks each consisting of a SOS-flow layer, a normalization flow layer and a reversing layer. The number of blocks is set to $1$. In the SOS-flow layer, we set $k = 5$ and $r = 4$ in Eqn. (8) in [22]. All the parameters are generated by a conditioner network, which contains one fully-connected layer: FC(724, 64, none)-FC(64, $c$, none), where $c$ is the number of the parameters in the SOS-flow layer.
    - Optimization hyperparameters: The number of epochs is set to $1000$, and training is stopped after 100 epochs of non-decreasing loss. The size of each mini-batch is $256$. We use Adam with the learning rate $10^{-5}$.

- **Fashion-MNIST** [57]
    - Dataset description: Fashion-MNIST includes $70,000$ grayscale images of fashion products in 10 classes. This dataset has the same image size and the structure of training and test splits as in MNIST.
    - Network architecture: same as that in MNIST.
    - Optimization hyperparameters: same as those in MNIST.

- **KDDCUP** [27]
    - Dataset description: KDDCUP dataset contains 125 dimensions in total. In this dataset, $20\%$ of data are labeled as "normal" and the rest are labeled as "attack". We treat "normal" data as novel since they are minority.
    - Network architecture: We use the same autoencoder as that in DAGMM except that the dimension of the latent vector is set to 2. The structure of the autoencdoer is as follows: FC(125,60,tanh)-FC(60,30,tanh)-FC(60,30,tanh)-FC(30,10,tanh)-FC(10,2,none)-FC(2,10,tanh)-FC(10,30,tanh)-FC(30,60,tanh)-FC(60,125,tanh).
    - Optimization hyperparameters: The size of each mini-batch is $1024$. The learning rate in Adam is $10^{-5}$. Training is stopped after 100 epochs of non-decreasing loss.

- **Thyroid** [28]
    - Dataset description: Thyroid dataset consists of three classes. We treat the hyperfunction class as the novel class and the rest as the nominal class.
    - Network architecture: We remove the autoencoder and only use the same estimation network as that in MNIST.
    - Optimization hyperparameters: The size of each mini-batch is $1024$. The learning rate in Adam is $10^{-3}$. Training is stopped after 100 epochs of non-decreasing loss.

Table 5: Statistics of Datasets

|  | Dimension | Instance | Classes | Anomaly ratio |
|---|---|---|---|---|
| MNIST | 784 | 70,000 | 10 | 0.9 |
| Fashion-MNIST | 784 | 70,000 | 10 | 0.9 |
| KDDCUP | 125 | 494,021 | 2 | 0.2 |
| Thyroid | 6 | 3,772 | 2 | 0.025 |

# B Comparison between density and quantile approaches

**Theorem 1** *In the univariate case, if the nominal distribution $F_0$ is unimodal and symmetric w.r.t the origin, then the density approach and the quantile approach achieve the same* AUROC.

*Proof:* It is well-known that AUROC is equal to the probability of a random nominal example being ranked higher than a random novel example, i.e.,

$$\text{AUROC} = \Pr(\mathsf{S}(X_0) > \mathsf{S}(X_1)),$$

where $X_0 \sim F_0$ is nominal and $X_1 \sim F_1$ is novel, and $\mathsf{S}$ is the scoring rule.

For the density approach, we have $\mathsf{S} = f_0$, where $f_0 = F_0'$ is the density of the nominal distribution. Thus,

$$\text{AUROC}_{\text{NLL}} = \Pr(f_0(X_0) > f_0(X_1)) = \Pr(|X_0| < |X_1|),$$

where the last equality follows from the unimodal and symmetric assumption on $f_0$.

On the other hand, for the quantile approach, the scoring rule is $\mathsf{S} = -|F_0 - \frac{1}{2}|$ (note the negation since we assume the higher $\mathsf{S}$ is the more nominal it is). Thus,

$$\begin{aligned}
\text{AUROC} &= \Pr(-|F_0(X_0) - \tfrac{1}{2}| > -|F_0(X_1) - \tfrac{1}{2}|) \\
&= \Pr(|F_0(X_0) - F_0(0)| < |F_0(X_1) - F_0(0)|) \\
&= \Pr(|X_0| < |X_1|),
\end{aligned}$$

where again the last equality is due to the unimodal and symmetric assumption on $F_0$. ∎

**Remark 2** *There is nothing special about the origin: the same result holds if $F_0$ is unimodal and symmetric w.r.t any point c.*

**Remark 3** *We suspect a similar result holds for multivariate distributions as well. A natural condition on $f_0$ is that its contours are multiples of the $\ell_\infty$ ball. We need to show that the TQM for such distributions are symmetric in some sense.*

**Theorem 2** *In the univariate case, if the nominal distribution $F_0$ is uni-modal and symmetric, then the density approach and the quantile approach lead to the same ROC curve.*

*Proof:* Denote the novel data as positive and nominal data as negative. For the quantile approach, given a threshold $t_q$ the set of data identified as novel can be characterized by $\{x : |F_0(x) - \frac{1}{2}| > t_q\}$. In contrast, for the density approach, given a threshold $t_d$ the identified novel data can be characterized by $\{x : -f_0(x) > t_d\}$, where $f_0(x)$ is the density of the nominal distribution. For both cases, the left hand side of the inequality represents the scoring function, and the higher the value of the scoring function the more likely the data being identified as novel.

In an ROC curve, each point is associated with a threshold. Therefore, to prove the result it suffices to show that there exists a one-to-one correspondence between $t_q$ and $t_d$ that leads to the same partition of the novel and nominal regions under the quantile and density approach respectively. Obviously, if $F_0$ is uni-model and symmetric, given $t_q$ we can set $t_d = -f_0(F^{-1}(t_q + \frac{1}{2}))$ and the partition is the same. ∎

**Remark 4** *In general the above conclusion cannot be extended to the multivariate case. For example, assume that the nominal data follows the 2-D standard Gaussian. Then, under the density approach, the boundary between novel and nominal data is an ellipsoid; while under the quantile approach, the boundary is square (assuming we employ the infinity norm scoring rule). The corresponding experimental results are shown below.*

## B.1 1-D: uni-model and symmetric model

Assume that the nominal data follows the standard univariate Gaussian distribution $N(10, 1)$. Consider two types of novel data: I) novel data are far away from nominal data, say following $N(15, 1)$;

Figure 2: Type I novel data. 1) distribution of test data; 2) distribution of pre-image in $[0,1]$ for nominal data; 3) distribution of pre-image in $[0,1]$ for novel data; and 4) ROC curve.

Figure 3: Type II novel data. 1) distribution of test data; 2) distribution of pre-image in $[0,1]$ for nominal data; 3) distribution of pre-image in $[0,1]$ for novel data; and 4) ROC curve.

and II) novel data are near nominal data, say following $N(12, 1)$. For both cases, the density and quantile methods have exactly the same ROC curve, confirming our theoretical results above. In particular, for the first case, the curve goes vertically from $(0,0)$ to $(0,1)$, and then horizontally to $(1,1)$, indicating perfect performance in anomaly detection. See Figures 2 and 3.

### B.2   1-D: mixture model

Assume that the nominal data follows the Gaussian mixture model $0.7N(0, 2^2) + 0.3N(10, 1)$. Consider two types of novel data: I) novel data is far away from nominal data, say following $N(15, 1)$; and II) novel data is surrounded by nominal data, say following $N(5, 1)$. For the first case, both methods have perfect performance; while for the second case, the quantile method is dominated by the density method. See Figures 4 and 5.

### B.3   2-D: uni-modal and symmetric model

Assume that the nominal data follows the 2-D Gaussian distribution with mean $[0, 0]$ and covariance matrix $[1, 0; 0, 1]$. Consider two types of novel data: I) novel data is far away from nominal data, say following the 2-D Gaussian distribution with mean $[5, 5]$ and covariance matrix $[1, 0; 0, 1]$; and II) novel data is near nominal data and follows the 2-D Gaussian distribution with mean $[2, 2]$ and covariance matrix $[1, 0; 0, 1]$. For the first case, both methods have perfect performance; while for the second case, the density method is slightly better than the quantile method. See Figures 6 and 7.

### B.4   2-D: donut example

Let us consider the donut distribution[2]

$$p(x, y) = \begin{cases} \frac{1}{3\pi}, & \text{if } 1 \leq x^2 + y^2 \leq 4 \\ 0, & \text{otherwise} \end{cases}.$$

Under the increasing triangular map $\mathbf{Q}$, the pre-images of $x$ and $y$ in $[0,1]^2$ are $F(x)$ and $F(y|x)$, respectively, where $F(\cdot)$ denotes the cumulative distribution function.

The marginal density of $x$ can be represented by

$$p(x) = \begin{cases} \frac{2}{3\pi}(\sqrt{4 - x^2} - \sqrt{1 - x^2}), & \text{if } -1 < x < 1 \\ \frac{2}{3\pi}\sqrt{4 - x^2}, & \text{if } -2 \leq x \leq -1 \text{ or } 1 \leq x \leq 2 \\ 0 & \text{otherwise} \end{cases}.$$

Figure 4: Type I novel data. 1) distribution of test data; 2) distribution of pre-image in $[0, 1]$ for nominal data; 3) distribution of pre-image in $[0, 1]$ for novel data; and 4) ROC curve.

Figure 5: Type II novel data. 1) distribution of test data; 2) distribution of pre-image in $[0, 1]$ for nominal data; 3) distribution of pre-image in $[0, 1]$ for novel data. The novel data are projected around the middle instead of at the ends; and 4) ROC curve.

Then $F(x)$ can be calculated as follows:

$$
F(x) = \begin{cases}
0, & \text{if } x < -2 \\
\frac{2}{3\pi}\left(\frac{x}{2}\sqrt{4-x^2} + 2\arcsin\frac{x}{2} + \pi\right), & \text{if } -2 \le x < -1 \\
\frac{2}{3\pi}\left(\frac{3}{4}\pi + \frac{x}{2}\sqrt{4-x^2} - \frac{x}{2}\sqrt{1-x^2} + 2\arcsin\frac{x}{2} - \frac{1}{2}\arcsin x\right), & \text{if } -1 \le x < 1 \\
\frac{2}{3\pi}\left(\frac{\pi}{2} + \frac{x}{2}\sqrt{4-x^2} + 2\arcsin\frac{x}{2}\right), & \text{if } 1 \le x < 2 \\
1, & \text{otherwise}
\end{cases}
$$

Given $x$, $y$ is uniformly distributed.

1. If $-2 \le x \le -1$ or $1 \le x \le 2$, the conditional density $p(y|x)$ can be represented by

$$
p(y|x) = \begin{cases}
\frac{1}{2\sqrt{4-x^2}}, & \text{if } -\sqrt{4-x^2} \le y \le \sqrt{4-x^2} \\
0, & \text{otherwise}
\end{cases},
$$

and the corresponding conditional CDF

$$
F(y|x) = \begin{cases}
0, & \text{if } y < -\sqrt{4-x^2} \\
\frac{y+\sqrt{4-x^2}}{2\sqrt{4-x^2}}, & \text{if } -\sqrt{4-x^2} \le y < \sqrt{4-x^2} \\
1, & \text{otherwise}
\end{cases}.
$$

2. If $-1 < x < 1$,

$$
p(y|x) = \begin{cases}
\frac{1}{2(\sqrt{4-x^2}-\sqrt{1-x^2})}, & \text{if } -\sqrt{4-x^2} \le y \le -\sqrt{1-x^2} \text{ or } \sqrt{1-x^2} \le y \le \sqrt{4-x^2} \\
0, & \text{otherwise}
\end{cases},
$$

and the corresponding conditional CDF

$$
F(y|x) = \begin{cases}
0, & \text{if } y < -\sqrt{4-x^2} \\
\frac{y+\sqrt{4-x^2}}{2(\sqrt{4-x^2}-\sqrt{1-x^2})}, & \text{if } -\sqrt{4-x^2} \le y < -\sqrt{1-x^2} \\
\frac{1}{2}, & \text{if } -\sqrt{1-x^2} \le y < \sqrt{1-x^2} \\
\frac{1}{2} + \frac{y-\sqrt{1-x^2}}{2(\sqrt{4-x^2}-\sqrt{1-x^2})}, & \text{if } \sqrt{1-x^2} \le y < \sqrt{4-x^2} \\
1, & \text{otherwise}
\end{cases}.
$$

On Figure 8 (left) we show the random samples of the nominal and novel data, and on Figure 8 (right) we show the pre-images of these samples in the square $[0, 1]^2$ using the derived analytical formula. It can be seen that the outer novel data is projected onto the boundary of the square hence can be

Figure 6: Type I novel data. 1) ROC curve; 2) density of nominal test data; 3) density of novel test data; 4) pre-image of nominal data in $[0,1]^2$; and 5) pre-image of novel data in $[0,1]^2$.

Figure 7: Type II novel data. 1) ROC curve; 2) density of nominal test data; 3) density of novel test data; 4) pre-image of nominal data in $[0,1]^2$; and 5) pre-image of novel data in $[0,1]^2$.

identified using $\texttt{TQM}_\infty$. The inner novel data, however, cannot be identified easily using the current quantile-based scoring functions. To improve the performance we might need some prior knowledge of such novel data and then adjust the scoring function accordingly. In contrast, the density approach would work well by setting a density threshold between $0$ and $\frac{1}{3\pi}$.

In Figure 9, instead of applying the analytical formula we use an SOS-based estimation network to learn the TQM. The observation is generally consistent with that derived using the analytical formula.

## B.5 Discussion

The quantile and density methods apply different scoring functions to identify novel data. Specifically, under the density method data with a low density (or log-likelihood) is deemed novel, while under the quantile method, for example, data projected to the boundary regions of the hypercube $[0,1]^d$ is deemed novel. A main advantage of the quantile method is that by checking whether data projected onto $[0,1]^d$ is uniformly distributed we can tell whether the quantile map is estimated successfully. In contrast, for the density method, generally it is difficult to assess the accuracy of the estimated density. To give an example, consider the donut example in Section B.4 and assume the outer data as novel. In Figure 10, we show the results when the TQM **Q** is learned by SOS and MAF [38], respectively. By projecting data onto $[0,1]^2$ (using the inverse TQM), we can conclude that SOS learns a better quantile map and indeed the corresponding ROC curve dominates that under MAF.

We also point out that under the current quantile thresholding rules (see §4.3) the identified nominal region is generally (path) connected, due to the increasing requirement we impose on TQM. Therefore, provided that nominal data follows some multi-modal distribution and novel data is located between different modes, as the shown example of 1-D mixture model in §B.2, the current quantile scoring rules would not work well. This reveals the importance of learning a (unimodal) hidden representation in our framework (2). It would be interesting to design new quantile thresholding rules to induce disconnected nominal region.

Figure 8: Donut example: 1) samples of nominal and two types of novel data; and 2) analytical pre-images in $[0, 1]^2$ for nominal and novel data.

Figure 9: Donut example: The density and the quantile map are learned by SOS. (1) pre-image of nominal training data in $[0, 1]^2$; 2) pre-image of test data in $[0, 1]^2$; 3) density of nominal training data; and 4) density of test data.

Figure 10: Donut example: In the first row, the density and the quantile map are learned by SOS: (1) pre-image of nominal training data in $[0, 1]^2$; 2) pre-image of test data in $[0, 1]^2$; and 3) density of test data. In the second row, the density and the quantile map are learned by MAF. The last plot shows the comparison of ROC curves.

## C  More results on comparison between joint and two-stage training

In Table 6, we show the comparison between joint and two-stage training on Fashion-MNIST dataset. The observation is similar to that on MNIST dataset.

Table 6: Comparison between joint and two-stage training: AUROC on Fashion-MNIST

| Class | Fix-NLL | Pretrain-NLL | Ours-NLL | Fix-$\mathrm{TQM}_2$ | Pretrain-$\mathrm{TQM}_2$ | Ours-$\mathrm{TQM}_2$ |
|---|---|---|---|---|---|---|
| 0 | 0.9114 | 0.8612 | **0.9217** | 0.8959 | 0.8650 | **0.9169** |
| 1 | 0.9764 | **0.9852** | 0.9579 | 0.9639 | **0.9813** | 0.9496 |
| 2 | 0.8799 | 0.8575 | **0.8985** | 0.8809 | 0.8548 | **0.8990** |
| 3 | **0.9370** | 0.9222 | 0.9304 | 0.9269 | 0.9233 | **0.9245** |
| 4 | 0.9013 | 0.9132 | **0.9223** | 0.8859 | 0.9080 | **0.9209** |
| 5 | 0.9096 | **0.9117** | 0.8940 | **0.9140** | 0.9098 | 0.8844 |
| 6 | 0.8424 | 0.7488 | **0.8435** | **0.8391** | 0.7617 | 0.8384 |
| 7 | 0.9757 | **0.9842** | 0.9802 | 0.9689 | **0.9843** | 0.9718 |
| 8 | 0.9125 | 0.8851 | **0.9450** | 0.8962 | 0.8750 | **0.9429** |
| 9 | 0.9776 | **0.9879** | 0.9825 | 0.9780 | 0.9827 | **0.9830** |
| avg | 0.9224 | 0.9057 | **0.9276** | 0.9150 | 0.9046 | **0.9234** |

## D  More results on visualization

In this section, we show more visualization results on the MNIST dataset. We use digit 1 as the nominal class. The results for other classes are similar. These visualizations can be used for diagnosing the training process and for assessing the quality of the learned TQM: by definition, the pre-image of data under TQM should be uniformly distributed on the hypercube $[0, 1]^m$.

Figure 11: All marginals of pre-image of training data in $[0, 1]^{64}$: 1) marginals at initialization; and 2) marginals at 1000 epochs of training.

Figure 12: All marginals of pre-image of test data in $[0, 1]^{64}$: 1) marginals at initialization; and 2) marginals at 1000 epochs of training.

Figure 13: Marginal and joint distributions of pre-image in $[0, 1]$ of training data (dimension: $56, 57,$ and $58$). 1) distributions at initialization; and 2) distributions at 1000 epochs of training.

Figure 14: Marginal and joint distributions of pre-image in $[0, 1]$ of test data (dimension: $56, 57,$ and $58$). 1) distributions at initialization; and 2) distributions at 1000 epochs of training.