[Reviews · NeurIPS 2019]

Reviewer 1



Similar to the univariate case, the ordering in the d-dimensional space should also be the constraint of the multivariate quantile function (MQF), correct? Would the parameterization of the sum-of-squares (SOS) flow enforce this constraint on the MQF? Besides SOS flow, is there any other generative model that can enforce this constraint? To demonstrate the advantage of the multiple gradient descent algorithm (MGDA) in optimizing the objective function (5), did the author run ablation study by comparing it against the baseline of training the reconstruction loss first and then optimizing the negative log-likelihood term? Is there any significant improvement by using MGDA except no need to tune lambda? The performance of difference approaches on the MNIST dataset are pretty close. Did the author run any statistical test to confirm the performance gap is significant?

Reviewer 2



I appreciated the author's response to my questions and felt they adequately answered the core issues. I have increased my score by one point. The proposal could be strengthened by showing a clear plausible (or even better real-world data) example of when the quantile approach is better. Currently, as a reader, I'm still not convinced that I would ever choose the quantile method over the likelihood method but I believe likely an example exists. ----------- Original review ---------- This paper proposes a novelty detection framework which is a synthesis of several components including feature extraction via neural networks, density estimation via flows and multiple gradient descent for optimization. The framework allows for novelty detection via two different methods: the first is based on quantiles and the other is based on likelihood. While each component is not novel, the synthesis of the components has some novelty. Overall, the paper was reasonably well-written. 1. Questionable novelty or core contribution of multivariate quantile function (MQF) definition The paper suggests that their definition of the MQF is novel and/or a core contribution when saying "We extend the ..." in the introduction and "We propose the following multivariate generalization of the quantile function..." Yet, even in the paper, it is mentioned that this definition was previously given in [9]. Also, this definition has similarities to the definition of a "density destructor" in [Inouye and Ravikumar, 2018] except that they define something akin to the CDF rather than inverse CDF (and only require invertibility rather than triangular mapping). Overall, the claim that this is a core contribution seems weak. What is novel or particularly interesting about this definition? Why should the definition of a multivariate extension of the quantile function require an increasing triangular map? Why not just require invertibility? (Maybe the definition could be named "Triangular Quantile Function" so that its meaning is obvious and clear rather than suggesting it is the most obvious extension of the univariate quantile function.) 2. Empirically, the results for log likelihood seem to always do similar or better than quantile-based (except for maybe the KDDCUP). One suggestion in the paper for using quantile is that it may be easier to set the threshold for quantile-based methods (maybe some more explanation for why this is actually easier would be good). Are there any other reasons to prefer the quantile method? For example, it seems that the likelihood method would do much better for a donut shaped distribution because the hole of the donut would be an outlier via NLL but not via quantiles. Why do you think it works better for the KDDCUP dataset versus the others? Adding a discussion on this simple case and others might be interesting. [Inouye and Ravikumar, 2018] Inouye, D. and Ravikumar, P. "Deep Density Destructors." ICML, 2018.

Reviewer 3



This paper presents a generic method for solving a practically relevant problem of novelty detection. The authors show that existing methods including the classical One-class SVM [42] as well as the recent Latent Space Autoregression [1] can be recovered as special cases of the proposed method. Overall the paper is well-written and ideas are presented clearly. On the technical front, it exploits the recent work [19] and combines the existing techniques to come up with a generic/unifying method for novelty detection. For this, it also introduces a novel multivariate generalization of quantile functions. What I further liked in the paper are 1. the proposed method allows one to define scoring rules for novelty detection based on both quantiles as well as the estimated density 2. instead of trying to find the right trade-off between reconstruction loss and the negative log-likelihood (via tuning the regularization parameter), it shows that one can employ multi-obejective optimization to get better results. 3. A thorough experiments are conducted by comparing several baseline methods with suitable scoring rules.

[Author Response · NeurIPS 2019]

We thank all reviewers for their critical comments and we address some questions below.

**Q:** (Reviewer 1) Ordering constraint on the multivariate quantile function
**A:** The reviewer is correct. Our definition of the quantile, similar to the SOS flow or other autoregressive models,
assumes that we have fixed an ordering of coordinates. This is not an issue here since we perform novelty detection in
the jointly learned latent space: choosing a different ordering simply amounts to permuting the latent space which will
not affect the end result. Yes, other neural density estimators, such as autoregressive models (MAF, IAF, MADE, NAF,
real NVP, or even the classic sigmoid belief networks), can also be plugged into our framework.

**Q:** (Reviewer 1) MGDA vs. two-stage training
**A:** Yes, we have tried the two-stage approach in our initial study. Similar as what many existing works reported [e.g. 1,
5, 39, 52], we found that it usually leads to suboptimal performance, possibly because the learned latent representation
may not be necessarily helpful for training the second SOS stage. MGDA is more robust and parameter-free.

**Q:** (Reviewer 1) Statistical test on MNIST results
**A:** We did not perform statistical test since some baselines only used the default train/test split on MNIST. Calculating
p-value based on 1 split does not seem to be meaningful. Nevertheless, we believe Table 3 suffices to show the
competitiveness of our algorithms. Our setup here also comply with existing works [e.g. 1, 5, 16, 34, 38, 39, 51, 52].

**Q:** (Reviewer 2) Novelty and core contribution with regard to MQF
**A:** We apologize for the confusion, and we will follow the reviewer's suggestion to tone down our contribution in this
definition and to adopt the more precise name "triangular quantile map" (TQM from now on).

In [9] Decurninge mentioned in passing that "For example, if we consider Rosenblatt transport," which is the only
sentence that hinted the TQM definition. In Decurninge's dissertation, he expanded the discussion on the "Rosenblatt
transport" which is essentially a constructive way to define (a version of) TQM. Decurninge's goal was to use TQM
as an intermediate tool to define L-moments, rather than treating it as an object of independent interest. Moreover,
our focus in this work, such as uniqueness, the importance of triangularity and monotonicity, an efficient estimation
algorithm, and the application to novelty detection, was never touched in Decurninge's work. We will follow the
reviewer's suggestion to tone down our role in this definition and discuss Decurninge's contribution in more details.

Thank you for bringing [Inouye and Ravikumar, 2018] to our attention! We will cite and discuss its relation to our
work. The major difference is that we insist the mapping to be triangular and monotonic while Inouye and Ravikumar
only require invertibility. Compared to Inouye and Ravikumar, TQM enjoys the following advantages: (a) Uniqueness.
The density destructor is not unique hence unidentifiable. For instance, consider the trivial task of destructing the
uniform density on $[0, 1]$: both $T(x) = x$ and $T(x) = 1 - x$ would do while the latter is not monotonic hence not
allowed in our definition. (In high dimensions the unidentifiability issue is even worse and it would be difficult to
define monotonicity in the absence of the triangular requirement.) (b) Computational efficiency. Inverting a monotonice
triangular map only takes a linear number of 1-d root findings while inverting the destructor of Inouye and Ravikumar
could be computationally challenging. Note that since we consider both the density and quantile rules, it is important to
compute both the map $Q$ and its inverse $Q^{-1}$. (c) Convenience. The triangular structure allows us to recycle many
existing neural density estimators, such as the SOS flow and autoregressive models (MAF, IAF, NAF, real NVP, etc.).

**Q:** (Reviewer 2) When is the quantile approach useful?
**A:** We want to emphasize that we do not claim the quantile rule is better than the density rule, or vice versa. Our point
is, through estimating the TQM, we now do not have to choose: both can be used at the same time. We have not tried to
optimize the quantile rule either: the ones we used in the experiments were designed for simplicity and to encourage
connectedness. On the other hand, thresholding the likelihood can create (many) highly disconnected (novel) regions,
which is more flexible but also prune to outliers. It would be interesting to explore the tradeoff and to combine the two
approaches in future work. We have run experiments on the donut example, see Figure 1. Note that if we change the
quantile rule slightly to $\alpha < \|\boldsymbol{\Phi}(\mathbf{T}^{-1}(\tilde{\mathbf{z}})) - \frac{1}{2}\|_\infty < \beta$ with suitable $\alpha$ and $\beta$ we can achieve similar performance as
the density approach (which has an intrinsic advantage on this example due to disconnectedness of the novel regions).
Note that since we perform novelty detection in the latent space $\mathbf{z}$, it is unlikely for the jointly learned latent space to be
disconnected (as in the donut example). Finally, we point out that TQM allows one to visualize the progress of the
estimation algorithm: the 1-d or 2-d projections of training samples onto coordinate axes should be uniform.

Figure 1: Donut example. From left to right: (1) uniform density over blue region; (2) pre-image in the cube using true
TQM; (3) pre-image of training sample using estimated TQM; (4) pre-image of test sample using estimated TQM; (5)
density of training sample using estimated TQM; (6) density of test sample using estimated TQM.

[Meta-Review · NeurIPS 2019]

This paper proposes a novelty detection framework, using feature extraction via neural networks, density estimation via flows, and multiple gradient descent for optimization. The reviewers were unanimous in their vote to accept. Authors are encouraged to revise with respect to reviewer comments.